# Soy Isoflavones Ameliorate Fatty Acid Metabolism of Visceral Adipose Tissue by Increasing the AMPK Activity in Male Rats with Diet-Induced Obesity (DIO)

**DOI:** 10.3390/molecules24152809

**Published:** 2019-08-01

**Authors:** Jinlong Tan, Chao Huang, Qihui Luo, Wentao Liu, Dongjing Cheng, Yifan Li, Yu Xia, Chao Li, Li Tang, Jing Fang, Kangcheng Pan, Yangping Ou, Anchun Cheng, Zhengli Chen

**Affiliations:** 1Laboratory of Animal Disease Model, College of Veterinary Medicine, Sichuan Agricultural University, Chengdu 611130, Sichuan, China; 2Key Laboratory of Animal Disease and Human Health of Sichuan Province, College of Veterinary Medicine, Sichuan Agricultural University, Chengdu 611130, Sichuan, China; 3College of Veterinary Medicine, Sichuan Agricultural University, Chengdu 611130, Sichuan, China

**Keywords:** soy isoflavones, AMPK, lipid homeostasis

## Abstract

Soy isoflavones are natural active ingredients of soy plants that are beneficial to many metabolic diseases, especially obesity. Many studies have reported that obesity is closely related to visceral fatty acid metabolism, but the effect has not been well defined. In this study, we show that soy isoflavones improve visceral fatty acid metabolism in diet-induced obese male rats, which was indicated by reduced body weight and visceral fat cell area, as well as suppressed visceral fat synthesis and accelerated fat hydrolysis. We also found that common components of soy isoflavones, daidzein and genistein, were able to inhibit the lipid accumulation process in 3T3-L1 cells. Moreover, we showed that soy isoflavones can promote on AMP-activated protein kinase (AMPK) activity both in vivo and in vitro, which may be implicated in lipid metabolism regulation of soy isoflavones. Our study demonstrates the potential of soy isoflavones as a mechanism for regulating lipid homeostasis in visceral adipose tissue, proven to be beneficial for obesity treatment.

## 1. Introduction

The epidemic of obesity has increasingly led to various diseases, including heart disease, diabetes, high blood pressure, etc. [1,2]. A substantial number of studies have reported that obesity is closely related to excess accumulation of visceral adipose tissue and causes inflammation, abnormal blood lipid levels, and insulin resistance, however, the pathogenesis of obesity still has not been well defined [3,4]. Visceral adipose tissue mainly includes kidney fat, epididymal fat, mesenteric fat, etc. [5]. It is involved in the synthesis, hydrolysis, and transport of free fatty acids, and an imbalance of this process causes excessive accumulation of visceral fat, leading to obesity [6,7,8,9,10].

The sterol regulatory element-binding protein family (SREBPs) is the key factor of transcription for lipogenesis, also involved in fat synthesis, and together with the hydrolases, adipose triglyceride lipase (ATGL), and hormone sensitive lipase (HSL) maintains fatty acid homeostasis [11,12]. AMP-activated protein kinase (AMPK) is a major regulator of lipid metabolism, plays a vital role in the regulation of cellular energy homeostasis, and serves as a potential therapeutic target for obesity [13]. AMPK has been widely reported to have beneficial effects in the liver and muscle. However, the role of AMPK in lipid metabolism of adipose tissue, an important component of metabolism, is not well defined [14,15]. As a heterotrimeric complex, AMPK consists of a catalytic alpha subunit, as well as regulatory beta and gamma subunits. The AMPK-α1 subunit, which plays a major role in adipose tissue and adipocytes, is involved in regulating the energy homeostasis of tissues and cells [16]. However, the exact mechanism of these processes remains unclear.

Soy is as an everyday essential in various Asian foods, and is especially in demand in China [17]. Soy isoflavones are natural phytoestrogens in soybeans. A substantial number of studies have reported that soy isoflavones have beneficial effects in diseases such as cardiovascular disease, hyperlipidemia, and osteoporosis [18,19]. In recent years, many studies have shown that the weak estrogen action of soy isoflavones is similar to that of estradiol and plays a role in lipid metabolism [20,21]. However, the exact mechanism of the role of soy isoflavones on lipid metabolism in male animals is not clear. In this study, we found that soy isoflavones can increase AMPK activity and reduce diet-induced obesity (DIO). Using soy isoflavones, visceral fat weight and lipid accumulation in diet-induced obese rats were reduced due to reduced visceral fat synthesis and accelerated hydrolysis. After soy isoflavones supplementation, phosphorylation of AMPK in visceral fat was enhanced, while SREBP-1 was inhibited, indicating that soy isoflavones increased AMPK activity and inhibited fat synthesis. Genistein and daidzein are common constituents of soy isoflavones. The results of in vitro experiments have shown that they increase AMPK activity in mature adipocytes and decrease lipid accumulation induced by dexamethasone (DEX), insulin, and 3-isobutyl-1-methylxanthine (IBMX). Taken together, these results demonstrate the potential of soy isoflavones in the treatment of obesity.

## 2. Results

### 2.1. Soy Isoflavones Reduced Body Weight and Hyperlipidemia in Male Rates with Diet-Induced Obesity (DIO)

The body weight and blood lipid levels of rats in each group were detected to investigate the effect of soy isoflavones on male rats with DIO. After consuming a high-fat diet for nine weeks, the rats weight was significantly higher than that of the rats that consumed the basal diet (Figure 1A), and the male rats with DIO had higher blood lipid levels (Figure 1B). The male rats with DIO were subsequently divided into an obese control group (OB) and low-, middle-, and high-dose soy isoflavone groups (LSI, MSI, HSI). The body weight and blood lipid levels of the soy isoflavone dose groups were significantly reduced after treatment with different doses of soy isoflavones as compared with rats in the OB group, especially in the middle- and high-dose soy isoflavone groups (Figure 1C,D). In addition, there was no significant difference in food intake between the rats in each dose group and the OB group (Figure 1E). These results show that soy isoflavones significantly ameliorate obesity in the OB male rats.

### 2.2. Soy Isoflavones Reduce Adipocyte Hypertrophy and Excessive Accumulation of Lipid in the Visceral Adipose Tissue of Male Rats with DIO

As the main area for energy storage, the visceral adipose tissue is always excessively accumulated in obese animals. As indicated in Figure 2A,B, the average area of a single visceral lipocyte in the male rats with DIO was significantly larger than those in the control group after feeding a high-fat diet. After the soy isoflavone treatment, the average area of a single visceral lipocyte was reduced significantly in all fat depots investigated as compared to rats with DIO. The weight of visceral fat also showed similar results (Figure 2C). Furthermore, the rats of the OB group showed an increased accumulation of lipids in visceral fat, and the lipid accumulation in rats with DIO was significantly reduced after the soy isoflavone treatment (Figure 2D). Daidzein and genistein, common components of soy isoflavones, were added to 3T3-L1 cells which did not affect cell viability (Figure 2E). Meanwhile, they significantly inhibited the lipid accumulation process induced by Dex, IBMX, and insulin (Figure 2F). These results reveal that soy isoflavones are involved in the lipid metabolism of adipocyte.

### 2.3. Soy Isoflavones Regulate Lipid Metabolism in the Visceral Adipose Tissue and Mature Adipocytes

Lipid metabolism of visceral fat involves the synthesis and hydrolysis of lipids. We detected the target genes associated with them. ACC1, ACC2, ACL, and FASN, enzymes related to lipogenesis are involved in fat synthesis. Meanwhile, SREBP-1c, a key transcription factor for lipogenesis, was also detected and involved in the regulation of lipase-related gene expression. In addition, ATGL and HSL are involved in the lipid hydrolysis process.

After feeding the high-fat diet, SREBP-1, the regulatory genes of lipid synthesis in obese rats, were significantly up-regulated as compared with the basal diet group. At the same time, the transcription level of its related enzyme gene was also significantly increased. Transcription of these genes was inhibited after the soy isoflavone treatment (Figure 3A). Western blots showed the same results as the gene expression (Figure 3B), and the high-fat diet increased the protein level of SREBP-1c in rat visceral adipose tissue. At the same time, the level of SREBP-1c protein was inhibited after the addition of soy isoflavones. The gene transcription levels of lipid hydrolysis were significantly reduced after feeding a high-fat diet as compared with the basal diet group. After the addition of soy isoflavones, the visceral fat hydrolysis in obese rats was enhanced (Figure 3C). Western blots showed the same results as the gene expressions. After feeding the high-fat diets, the protein levels of ATGL in various visceral adipose tissues of obese rats were significantly lower than those in basal diet rats. After the addition of soy isoflavones, the protein expression of ATGL was up-regulated (Figure 3D). Moreover, in vitro experiments showed the same results as in vivo. After the treatment of daidzein or genistein, the transcriptional level and the protein level of SREBP-1c were significantly down-regulated and the transcriptional level and protein level of ATGL were significantly up-regulated in adipocytes (Figure 3E). These results reveal an important role of soy isoflavones in regulating the lipids synthesis and hydrolysis of visceral fatty. However, more work is needed to explain how soy isoflavones regulate the above-mentioned process.

### 2.4. Soy Isoflavones Enhance the Activity of AMPK

To understand the mechanism by which soy isoflavones affect lipid metabolism in the visceral fat of obese rats, we measured the effect of soy isoflavones on AMPK activation on visceral adipose tissues. AMPK, a cellular energy sensor, plays a key role in the regulation of cellular energy homeostasis, and has a negative regulation effect on biosynthesis processes such as lipid synthesis that consume ATP. Phosphorylation of the catalytic subunit AMPKα at the threonine 172 site was detected, which is essential for AMPK activation. As shown in Figure 4A, the high-fat diet suppressed the protein levels of phosphor-AMPK (Thr172) in male rats and the addition of soy isoflavones significantly increased the protein levels of phosphor-AMPK (Thr172), which means that soy isoflavones could increase AMPK activity. In addition, the protein levels of phosphor-AMPK (172) were significantly increased after the addition of daidzein and genistein to 3T3-L1 cells that induced lipid accumulation (Figure 4B). These results indicate the promoting function of soy isoflavones on AMPK activity. AMPK-inhibitor was added to further verify that soy isoflavones regulate fatty acid metabolism via the AMPK/SREBP-1 pathway (Figure 4C). We found that the protein level of phosphor-AMPK (Thr-172) was reduced and the protein level of SREBP-1c was increased after adding Compound C, an AMPK-inhibitor. Conversely, the protein level of phosphor-AMPK (Thr-172) was increased and the protein level of SREBP-1c was reduced after the simultaneous addition of Compound C and daidzein or Compound C and genistein. Adding only daidzein and genistein showed a higher phosphor-AMPK (Thr-172) protein level and a lower SREBP-1c protein level. These results show that soy isoflavones regulate lipid metabolism in adipocytes through the AMPK/SREBP-1 pathway.

## 3. Discussion

Obesity is commonly seen in the twenty-first century, not only increasing body weight and excess visceral fat, which are two typical signs, but also increasing the risk of several serious chronic diseases such as hypercholesterolemia, hypertriglyceridemia, and cardiovascular disease [22,23]. Increased weight and excess visceral fat accumulation are typical signs of obese patients. Soy isoflavones, which are structurally similar to estradiol, were sparingly reported on the specific progress of obesity, especially in males [24]. The present study shows that soy isoflavones could ameliorate diet-induced obesity in male rats by reducing body weight, plasma lipid levels, and visceral fat area, consistent with most previous findings [25,26]. Furthermore, there are also few studies suggesting that the consumption of soy isoflavones could lead to weight gain [27]. Recently, studies conducted on the anti-obesity effects of soy isoflavones from molecular mechanisms and metabolic analysis [28], have reported that soy isoflavones increased the production of the metabolites, S-equol and O-desmethylangolensin (O-DMA), and could regulate various transcription factors that regulate metabolism such as PPARα, PPARβ, and PPARγ. Our work shows that overweight, excessive total cholesterol levels, triglyceride levels, and low-density lipoprotein levels were improved in rats with DIO after consumption of soy isoflavones. These results indicate the potential of soy isoflavones as an active ingredient in drugs to remediate obesity.

Adipocytes act like a warehouse to synthesize and store fat. A considerable number of studies have reported that there was a larger area of single lipocyte in obese individuals as compared with a normal individual [29,30,31]. In the present study, the average area of a single lipocyte in various visceral adipose tissues of rats with DIO was larger than that of rats in the basal diet group, which is consistent with previous studies. Moreover, we found that soy isoflavones reduced lipid accumulation in various visceral adipose tissues of rats with DIO. In addition, adding daidzein and genistein, common components of soy isoflavones, could reduce lipid accumulation in differentiated 3T3-L1 cells induced by dexamethasone, IBMX, and insulin. These results reveal that the lipid-lowering effects of soy isoflavones and consumption of soy isoflavones are involved in the lipid metabolism of visceral adipose tissue.

An imbalance of lipid homeostasis could cause disorders in fatty acid metabolism and then obesity. This metabolic mechanism involves the synthesis and hydrolysis of fatty acids. ACC1, ACC2, ACL, and FASN play a role in lipid synthesis and are regulated by the transcription factor SREBP-1. In a previous study, soy isoflavones were shown to inhibit SREBP-1 expression and fatty liver formation in the liver of rats with DIO [32]. We found that soy isoflavones inhibited the expression of SREBP-1c in visceral adipose tissue, which may indicate a prevalence of inhibition of soy isoflavones on lipid synthesis sites. Moreover, ATGL and HSL, two major lipid hydrolases, were found to increase expression in visceral adipose tissue of rats with DIO after consumption of soy isoflavones as compared with the rats fed a high-fat diet. Meanwhile, in vitro experiments also showed the same result that the addition of daidzein and genistein in the differentiated 3T3-L1 cells, i.e., SREBP-1c was inhibited and ATGL was up-regulated. In vivo and in vitro experiments revealed that soy isoflavones not only inhibited lipid synthesis but also promoted lipid hydrolysis. This conclusion suggests a two-way effect of soy isoflavones on lipid inhibition and hydrolysis.

AMPK, which is a key regulator in the regulation of cellular and systemic energy balance, is considered to be a target for the treatment of obesity. Although the majority of studies on AMPK focus on liver and skeletal muscle, more and more literature has confirmed the important role of AMPK in the metabolism of adipose tissue [33,34,35,36,37]. Furthermore, it has been shown that dietary phytoestrogens improve lipid metabolism via activating AMPK in other publications. Cederroth et al. reported that soy isoflavones promote the transfer of triglycerides to the liver by increasing AMPK activity [38]. Hwang et al. reported that β-sitosterol inhibits ACC phosphorylation by increasing AMPK activity to achieve lipid reduction [39]. Our research focuses on the purification of soy isoflavone phytoestrogens, genistein and daidzein, and whether to activate AMPK to regulate the key transcription factor for lipogenesis, SREBP-1. The subtype AMPKα is the major subtype of AMPK expressed in adipose tissue and is involved in the regulation of fat energy homeostasis [40]. A growing number of studies have reported on the lipid regulation process mediated by the AMPK/SREBP1 signaling pathway and suggests that phosphorylation of AMPK inhibits the expression of SREBP1 to reduce lipogenesis [41,42]. In contrast, in our study, rats with DIO that consumed soy isoflavones expressed higher levels of phosphor-AMPK (Thr172) and lower levels of SREBP1c as compared with the OB group rats, suggesting that the inhibitory effect of soy isoflavones may be regulated by the AMPK/SREBP1c pathway. After adding Compound C, an AMPK-inhibitor, the protein level of phosphor-AMPK (Thr-172) was reduced and the protein level of SREBP-1c was increased. Conversely, the protein level of phosphor-AMPK (Thr-172) was increased and the protein level of SREBP-1c was reduced after the simultaneous addition of Compound C and daidzein or Compound C and genistein. Only adding daidzein and genistein showed higher levels of phosphor-AMPK (Thr-172) and lower levels of SREBP1c. These results indicate soy isoflavones regulate lipid metabolism in adipocyte through the AMPK/SREBP-1 pathway.

## 4. Materials and Methods

### 4.1. Animal Care and Maintenance

All animal experiments were performed in accordance with the Animal Care and Use Committee Guidelines of the Sichuan Agricultural University, China. After one week of adaptation in a nonspecific pathogen environment, 100 male Sprague-Dawley (SD) rats were randomized into a basal diet group (Table 1) (Dashuo, Chengdu, China; control, Ctr, n = 16) and a high-fat diet group (Table 2) (n = 64). These rats were treated with the indicated diets for nine weeks to induce obesity, and the body weight was measured weekly. The criterion for the rats with DIO was that the body weight of the rats in the high-fat diet group was 20% more than that of the control group. The rats with DIO were further randomly divided into four groups (n = 16/group), and they were fed a high-fat diet with different doses of soy isoflavone additions (soy isoflavone extracts, North China Pharmaceutical Co., Ltd., Shijiazhuang, China) for 4 weeks, as described in Table 3. The compounds of soy isoflavone extracts, as quantified by HPLC, are shown in Table 4. After the end of the treatment, the rats were decapitated and the visceral fat (mesenteric, epididymal, and perirenal) was collected. The epididymal perirenal fat was weighed and parts of visceral fat were snap frozen in liquid nitrogen and stored at −80 °C for making frozen sections and determination of mRNA or protein levels. Part of the visceral fat was removed and fixed in 4% paraformaldehyde in preparation for hematoxylin and eosin staining. All protocols were approved by the Animal Care and Use committee of Sichuan Agricultural University, following the guidelines on animal experiments which come under the permit No. DY-S20174411, date: 10th January, 2018.

### 4.2. Body Weight, Fat Weight, Food Intake, and Plasma Measurement Fat Weight

Body weight and food intake were measured weekly. A blood sample was collected from the lateral tail vein. A 1 mm to 2 mm section was cut in the tip of the tail with a sterile scalpel blade. Blood was then milked from the base of the tail to the tip until a sufficient volume of blood was collected. These were collected for blood biochemical analysis (Beckman CX4, Indianapolis, IN, USA) at the ninth and thirteenth week. After the rats were sacrificed, epididymal fat, perirenal fat, and mesenteric fat were taken, and the weight of the epididymal and perirenal fat (the mesenteric fat containing lymph nodes was not weighed) was measured.

### 4.3. Histopathologic Evaluation

Part of the adipose tissue was fixed with 4% paraformaldehyde and then used to make paraffin sections. Next, hematoxylin and eosin (H and E) staining was performed according to the manufacturer’s instructions (Servicebio, Wuhan, China) and the average area of individual adipocytes was measured using Image Pro Plus 6.0 (Media Cybernetics, Bethesda, MD, USA) software.

To detect lipid accumulation in the adipose tissue and 3T3-L1 adipocytes, the adipose tissue frozen sections (10 μM) and the 3T3-L1 adipocytes were grown on cover glass and rinsed with distilled water. The Oil red O stock solution (Servicebio) was mixed with distilled water (6:4) and further filtered to obtain a fresh Oil red O solution. The sections were washed twice with phosphate buffered saline (PBS) and fixed with 10% formaldehyde for 1 h and then washed with PBS, washed with 60% isopropanol and dried. The Oil red O working solution was added and incubated for 30 min. It was then washed 3 times with distilled water. The sections were mounted with glycerinated gelatin and photographed using a microscope (Nikon, Japan).

### 4.4. Cell Culture, Differentiation, and Treatment

The 3T3-L1 cells were routinely cultured in Dulbecco’s modified Eagle’s medium (DMEM) (Gibco, Waltham, MA, USA) supplemented with 10% fetal bovine serum (FBS) in an incubator under an atmosphere of 5% CO_2_ at 37 °C and cultured for 2 days until the cells grew to about 80%. The cells were induced to differentiate by incubation in a differentiation induction medium consisting of DMEM, 10% FBS, 10 mg/L insulin, 1 µM dexamethasone (DEX), and 0.5 mM 3-isobutyl-1-methylxanthine (IBMX) for 2 days. The culture medium was then replaced with differentiation maintenance medium, consisting of DMEM, 10 mg/L insulin, and 10% FBS for 5 days, and the medium was changed once on day 2. Then the cells were harvested for the experiment.

To evaluate the effect of soy isoflavones on lipid accumulation and its suppression function, 25 μM of daidzein (Sigma-Aldrich, St. Louis, MO, USA) and 25 μM of genistein (Sigma-Aldrich) were added to the induction medium and differentiation medium. In the AMPK inhibition assay, 5 μM of Compound C (Selleck Chemicals, Houston, TX, USA) were added simultaneously with daidzein or genistein.

### 4.5. Cell Viability Assay

The 3T3-L1 cells were plated on 96-well plates. After the daidzein and genistein treatment, cell viability was measured using a Cell Counting Kit-8 (CCK-8) system (Solarbio, Peking, China) according to the manufacturer’s instructions. Briefly, 10 μL of CCK-8 solution was added to each well and the plates were incubated for 1–4 h. The absorbance at 450 nm was measured with a microplate reader (Thermo, Waltham, MA, USA). The cell viability assay was calculated as follows: cell viability assay (%) = [(experimental optical density (OD) value − blank control OD value)/(normal control OD value − blank control OD value)] × 100%.

### 4.6. Quantitative Real-Time Reverse-Transcription Polymerase Chain Reaction (Q-RT-PCR)

Total RNA was isolated from the epididymal, perirenal, and mesenteric adipose tissue, and adipocytes using RNAiso Plus (TaKaRa, Dalian, China). Total RNA was subjected to reverse transcription using a PrimeScript RT reagent Kit with gDNA Eraser (Perfect Real Time) (TaKaRa). Q-RT-PCR was performed using the Bio-Rad iQ5 system, and the relative gene expression was normalized to internal control as β-Actin. Primers for genes used in this study are shown in Table 5.

### 4.7. Western Blotting

The standard procedure for Western blotting was performed using the following antibodies: anti-adipose triglyceride lipase antibody (abcam, ab109251, Cambridge, MA, USA), anti-AMPKα antibody (CST, mAb #2603, Danvers, MA, USA), anti-phospho-AMPKα antibody (CST, mAb #2535), anti-SREBP1 antibody (Novus Biologicals, NB-100-2215SS, Littleton, CO, USA), anti-β-actin antibody (Boster, BM0627, Wuhan, China). Western blot analysis was done as described elsewhere [43]. Briefly, whole-tissue and cell lysates were prepared, and protein concentrations were measured using the bicinchoninic acid (BCA) method. Proteins were separated on the Bio-Rad Mini-PROTEAN Tetra gel system (BioRad, Hercules, CA, USA). The proteins were transferred to PVDF membranes and probed. Equal loading of the protein was confirmed by β-actin.

### 4.8. Statistical Analysis

One-way ANOVA and post hoc tests were performed with the Statistical Program for Social Sciences software for Windows (SPSS, version 17.0; SPSS Inc, Chicago, IL, USA). The data represent the mean and standard error of the mean (SEM). * p < 0.05, ** p < 0.01.

## Figures and Tables

**Figure 1 molecules-24-02809-f001:**
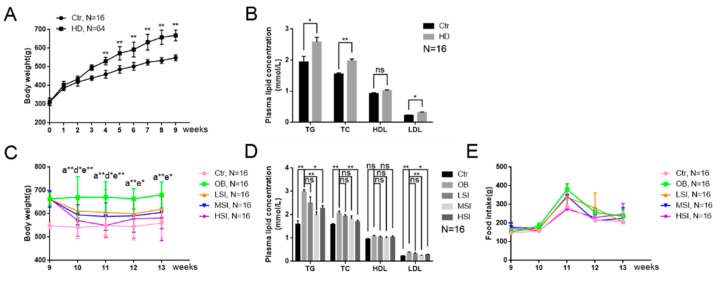
Soy isoflavones reduce body weight and the serum triglycerides (TG), total cholesterol (TC), and low-density lipoprotein (LDL) concentrations. (**A**) The body weight of rats fed a basal diet and the high-fat diet for nine weeks; (**B**) the plasma TG, TC, high-density lipoprotein (HDL), and low-density lipoprotein (LDL) concentrations of rats fed a basal diet and high-fat diet; (**C**) the body weight trend of rats in the control, obesity control group (OB), low dose of isoflavones (LSI), middle dose of isoflavones (MSI), and high dose of isoflavones (HSI) groups. a**: b vs. a, p < 0.01; d*: b vs. d, p < 0.05; e*: b vs. e, p < 0.05; e**: b vs. e, p < 0.01; (**D**) the plasma TG, TC, HDL, and LDL concentration of rats in each group at the thirteenth week; and (**E**) feed intake of rats in each group during the intervention of soy isoflavones. Error bars indicate SEM. * p < 0.05, ** p < 0.01. ns, no statistical significance. ^a^ control group, ^b^ obesity group, ^c^ low-dose soy isoflavones group, ^d^ middle-dose soy isoflavones group, ^e^ high-dose soy isoflavones group.

**Figure 2 molecules-24-02809-f002:**
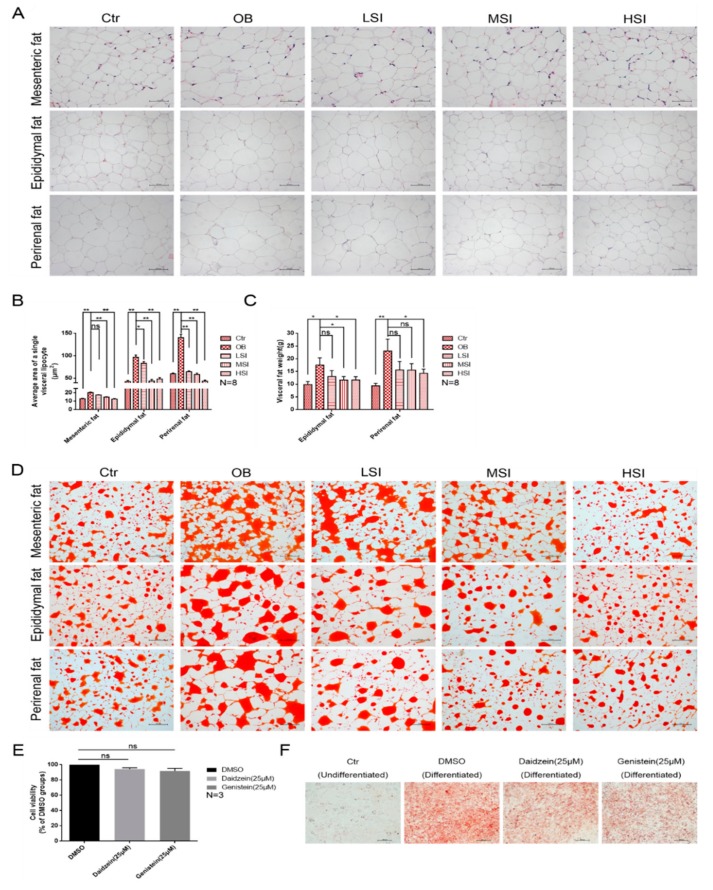
Soy isoflavones reduce the average area of a single visceral lipocyte and lipid accumulation. (**A**) Representative images of H and E staining in visceral adipose tissues, bar: mesenteric fat 50 µM, epididymal fat 100 µM, and perirenal fat 100 µM; (**B**) the average area of a single visceral lipocyte in each group rats; (**C**) the weight of visceral adipose tissues; (**D**) representative images of Oil red O staining in visceral adipose tissues, bar: 200 µM; (**E**) the effect of daidzein and genistein on cell viability; and (**F**) representative images of Oil red O staining in each group 3T3-L1 cells, bar: 100 µM.

**Figure 3 molecules-24-02809-f003:**
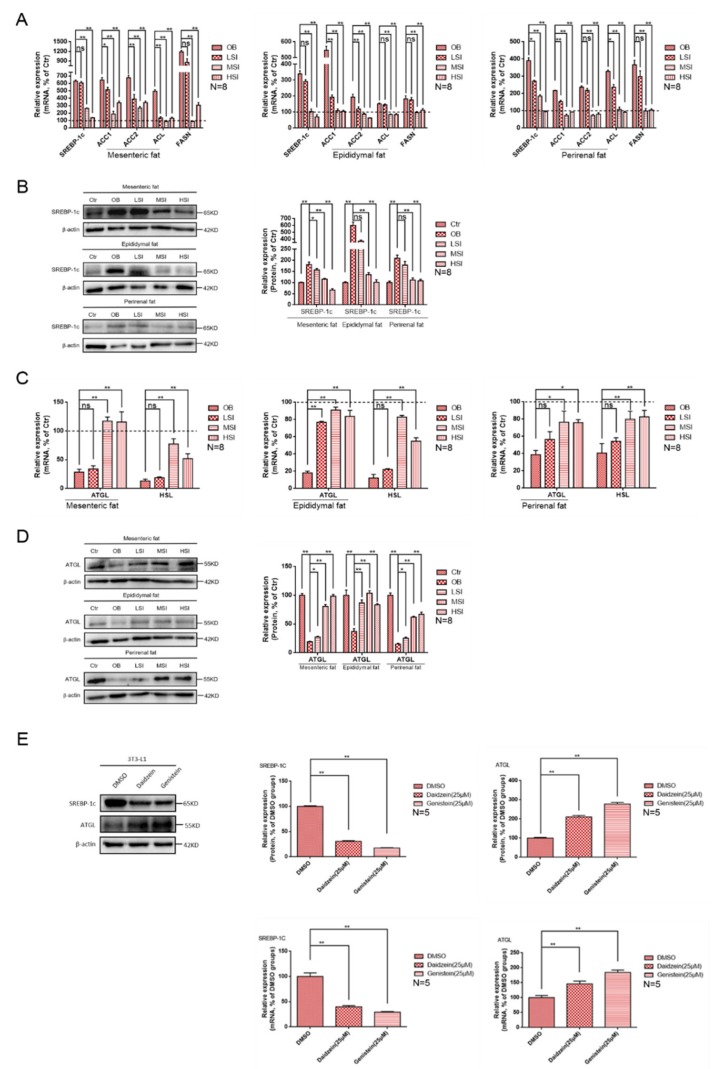
Soy isoflavones promote fatty acid hydrolysis and inhibit fatty acid synthesis. (**A**) Relative mRNA expression of visceral fat synthesis-related genes in each group of rats, the dotted lines in the graphs represent the control group; (**B**) the western blot results of sterol regulatory element-binding transcription factor 1c (SREBP-1c) in visceral fat of each group of rats; (**C**) relative mRNA expression of visceral fat hydrolysis-related genes in each group of rats, the dotted lines in the graphs represent the control group; (**D**) the western blot results of adipose triglyceride lipase (ATGL) in visceral fat of each group of rats; and (**E**) effects of daidzein and genistein on the mRNA levels and the protein levels of ATGL and SREBP-1c in differentiated 3T3-L1 cells.

**Figure 4 molecules-24-02809-f004:**
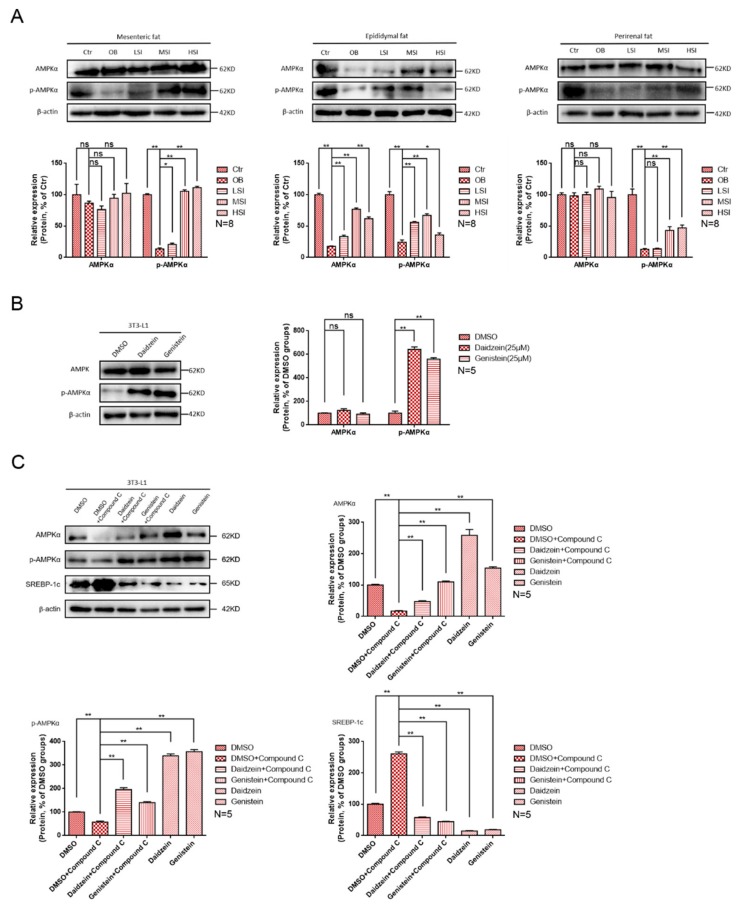
Soy isoflavones increase AMP-activated protein kinase (AMPK) activity in visceral fat and 3T3-L1 cells. (**A**) The western blot results of AMPKα and phosphor-AMPKα in visceral fat of each group of rats; (**B**) the western blot results of AMPKα and phosphor-AMPKα in differentiated 3T3-L1 cells after treatment with daidzein and genistein; and (**C**) the western blot results of AMPKα, phosphor-AMPKα, and SREBP-1c in differentiated 3T3-L1 cells after treatment with Compound C, daidzein, and genistein.

**Table 1 molecules-24-02809-t001:** Composition of the basal diet.

Ingredients	Content
Corn	54.0%
Fish meal	6.0%
Wheat bran	14.0%
Alfalfa meal	13.0%
Cotton meal	10.0%
Limestone	1.00%
Dicalcium phosphate	0.2%
Sodium chloride	0.3%
Vitamin and mineral	1.5%

**Table 2 molecules-24-02809-t002:** Composition of the high-fat diets.

Ingredients	Content
Basal diet	69.5%
Pork fat	15%
Sucrose	15%
Pig bile	0.5%

**Table 3 molecules-24-02809-t003:** Composition of the experimental diets from week 9 to week 13.

Groups	Control (Ctr, n = 16)	Obesity (OB, n = 16)	Low-Dose Soy Isoflavones (LSI, n = 16)	Middle-Dose Soy Isoflavones (MSI, n = 16)	High-Dose Soy Isoflavones (HSI, n = 16)
Diets	Basal diet	High-fat diet	High-fat diet + 50 mg/kg SIF	High-fat diet + 150 mg/kg SIF	High-fat diet + 450 mg/kg SIF

**Table 4 molecules-24-02809-t004:** Composition of the soy isoflavone extracts.

Compounds	Content
Daidzin	50.98%
Glycitin	30.36%
Genistein	8.80%
Daidzein	1.24%
Genistin	0.06%
Total isoflavones (HPLC)	91.64%

**Table 5 molecules-24-02809-t005:** Primers used for the real-time PCR analysis.

Gene	Primer Sequence (5′-3′)
*ACC1*	F: ATTGTGGCTCAAACTGCAGGT
	R: GCCAATCCACTCGAAGACCA
*ACC2*	F: CAACATCCGTCAGACGACCTC
	R: CGGACTCGTTGGTGATGAAGA
*ACL*	F: GCAGCACGTGATCCATGAAT
	R: GTGGGATGCTGGACAACATC
*FASN*	F: GCATTTCCACAACCCCAACC
	R: AACGAGTTGATGCCCACGAT
*SREBP-1c*	F: TGGACTACTAGTGTTGGCCTGCTT
	R: ATCCAGGTCAGCTTGTTTGCGATG
*ATGL*	F: TGGCGGCATTTCAGACAACT
	R: GTCCATCTCGGTAGCCCTGTT
*HSL*	F: GCGGACCAGCTCTAAAGAAAGA
	R: TTTCATCCTTCTGCCCCCTAC
*AMPK*	F: TTCGGGAAAGTGAAGGTGGG
	R: GGTTCTGGATCTCTCTGCGG
*β-actin*	F: ACGGTCAGGTCATCACTATCG
	R: GGCATAGAGGTCTTTACGGATG

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
