# Peer review of "Soy Isoflavones Ameliorate Fatty Acid Metabolism of Visceral Adipose Tissue by Increasing the AMPK Activity in Male Rats with Diet-Induced Obesity (DIO)"

_molecules, 2019, doi:10.3390/molecules24152809_

Round 1

Reviewer 1 Report

In this manuscript, Tan et al. report the effects of soy isoflavones on lipid metabolism in a rat model of diet-induced obesity. In particular, they claim that the beneficial effects are mediated by AMPK in visceral adipose tissue.

While the paper is stringently written and the design of the study is appropriate, the novelty of the results in regard of already published articles is not clear to me. Moreover, I strongly recommend that the manuscript is edited for correct English language – this will significantly improve the quality of the manuscript.

In the following, I listed a number of major and minor points, which in my eyes should be addressed in a revised version of the manuscript.

Major points:

1.      While AMPK is an important regulator of energy homeostasis, I am not convinced that it is the only player mediating the effects of soy isoflavones in adipose tissue. The same group recently published in the same journal that soy isoflavones regulated lipid metabolism in the liver via Akt/mTOR pathway. Have they checked this pathway also in this study? To further prove that AMPK is relevant in this context they could also use an AMPK inhibitor in vitro to show that this would inhibit the effects of the treatment.

2.      Soy isoflavones have been shown to improve lipid metabolism via AMPK in other publications (e.g. Cederroth et al. Diabetes 2008, Hwang et al. BBRC 2008). What is the novelty in their findings? Maybe this can be more pronounced in the discussion.

3.      Introduction, page 2: “…, the role of AMPK in adipose tissue, an important component of metabolism, is unclear.” Please be more precise – unclear in which aspect?

4.      Intro page 2 “…, the exact mechanisms of soy isoflavones on lipid metabolism in male animals is not clear.” Why do the authors pronounce male animals? Is there a known gender differences regarding the effect of soy isoflavones?

5.      Figures in general: Information on the number of independent experiments or significance thresholds are missing in several figures.

6.      Fig. 1C: labelling of group significance values is not intuitive

7.      Fig. 2D: The quality of the ORO stainings are not sufficient as most of the adipocytes are not stained at all. I would be easier to show triglyceride content of the tissues instead.

8.      Fig. 3C: The control group is missing.

Minor points (I only correct major typos and inconsistencies – editing strongly recommended):

1.      Abstract page 1 line 27 might read “We also found that common components…”

2.      page 1 line 29: “further shown…”

3.      2.2 it is not clear form the text how cell viability was measured

4.      page 3 line 99 might read: “…in all fat depots investigated as compared to DIO rats.”

5.      Fig 2 legend: there are two bar sized given – which one is correct?

6.      Fig 3B: check legends: bands are either labelled with b-actin or the adipose tissue. Please label the band with the protein name.

7.      page 7 line 151: “…homeostasis, and has a negative…”

8.      Fig 4 legend: use “treatment” instead of “intervention”

9.      page 7, lines 169/170: check grammar

10.  page 6 line 194: should read SREP-1?

11.  page 8 line 194: There is only one study cites, not studies

table 1: Sodium chloride

Reviewer 2 Report

The authors describe how the isoflavones improve visceral fatty acid metabolism in an obese rat model showing results in three different visceral fat pads: mesenteric, perirenal and epididymal; and then, they also show in vitro results, in 3T3-L1 cells. In general the manuscript is sound, the topic of this manuscript is of general interest and the results support the conclusions. Just some minor comments that I would like to suggest:

1-Figue 2C: capital letter in “perirenal fat”.

2-Figure 3E: put the western panel side by side with the graph results, no in the RNA graphs.

3-Figure 3A and 3C: indicate in the legend that the discontinue line in the graphs represent the controls.

4-Material and methods: in western blot section, indicate some references about how the procedure is done.

Author Response

Point 1: Figue 2C: capital letter in “perirenal fat”.

Response 1: We appreciate the reviewer’s concern, and we have made revision.

Point 2: Figure 3E: put the western panel side by side with the graph results, no in the RNA graphs.

Response 2: We appreciate the reviewer’s concern, and we have made revision.

Point 3: Figure 3A and 3C: indicate in the legend that the discontinue line in the graphs represent the controls.

Response 3: We appreciate the reviewer’s concern, and we have made revision.

Point 4: Material and methods: in western blot section, indicate some references about how the procedure is done.

Response 4: We appreciate the reviewer’s concern, we have indicated the relative reference about how the procedure is done.

Reviewer 3 Report

This manuscript was objected to investigate the effect of soy isoflavone on fatty acid metabolism in diet induced obesity rats. This study seems to be interesting cause most studies for soy isoflavones have been focused on OVA rat model, but this study deal with diet induced obesity model. The study design was good and results were well described.

Before accepting this study, the method for preparation of isoflavone samples should be added. In addition, in the discussion part, the originality should be more emphasized.

Author Response

Point 1: Before accepting this study, the method for preparation of isoflavone samples should be added. In addition, in the discussion part, the originality should be more emphasized.

Response 1: We appreciate the reviewer’s concern, the soy isoflavones used by us are commercial reagents. We are unable to obtain a preparation method of soy isoflavones prepared by the manufacturer, but we provide the ingredients of the commercial isoflavones in Table 4. Furthermore, we will emphasize originality in the discussion.

Round 2

Reviewer 1 Report

The quality of the manuscript significantly improved, all of my Point have been addressed.

Still, I would strongly recommend editing of English language.